# The prevalence of alcohol consumption and its related factors in adolescents: Findings from Global School-based Student Health Survey

Vahid Farnia[1], Touraj Ahmadi Jouybari[1], Safora Salemi[1]*, Mehdi Moradinazar[1], Fatemeh Khosravi Shadmani[2], Bahareh Rahami[1], Mostafa Alikhani[1], Shahab Bahadorinia[1], Tahereh Mohammadi Majd[3]

1 Substance Abuse Prevention Research Center, Health Institute, Kermanshah University of Medical Sciences, Kermanshah, Iran, 2 Research Center for Environmental Determinants of Health (RCEDH), Health Institute, Kermanshah University of Medical Sciences, Kermanshah, Iran, 3 Department of Biostatistics, School of Health, Kermanshah University of Medical Sciences, Kermanshah, Iran

* safora.salemi@kums.ac.ir

**Data Availability Statement:** The data are available on www.who.int/chp/gshs/factsheets/en.

## Abstract

### Background

Alcohol consumption has become very common among adolescents in recent years and its prevalence varies in different countries. This study aimed to investigate the prevalence of alcohol consumption and related factors in adolescents aged 11 to 16 years.

### Methods

This descriptive cross-sectional study was performed on 288385 adolescents (girls, 53.9% of total) aged 11 to 16 years. In the present study, the GSHS data (2003–2018) available to public on the websites of the US Centers for Disease Control and Prevention (CDC) and WHO was used. To investigate the factors affecting alcohol consumption, univariate and multivariate logistics models with 95% confidence limits were used.

### Results

The overall prevalence of alcohol consumption in adolescents was 25.2%, which was 28.3% and 22.4% in boys and girls, respectively. Among the surveyed countries, the highest prevalence was in Seychelles (57.9%) and the lowest in Tajikistan (0.7). Multivariate analysis showed that the Age for 16 and more than 16 years old (OR = 3.08,95%CI: 2.54–3.74), truancy for more than 10 days (OR = 1.24, 95%CI: 1.08–1.43), loneliness at sometimes of the times (OR = 1.04, 95%CI: 1.01–1.07), insomnia at most of the times (OR = 1.85, 95%CI: 1.70–2.01), daily activity (OR = 1.03, 95%CI: 1.00–1.07), bullied for 1–9 Days in a month (OR = 1.24, 95%CI: 1.09–1.40), cigarette (OR = 4.01, 95%CI: 3.86–4.17), used marijuana for more than 10 days in a month (OR = 5.58, 95%CI: 4.59–7.78), had sex (OR = 2.76, 95% CI: 2.68–2.84), and suicide plan (OR = 1.48, 95%CI: 1.42–1.54) were important factors

**Funding:** The Research Deputy of Kermanshah University of Medical Sciences funded the study (award number:980644). The funders had no role in study design, data collection and analysis, decision to publish, or preparation of the manuscript.

**Competing interests:** The authors have declared that no competing interests exist.

**Abbreviations:** GSHS, Global School-based Student Health Survey; WHO, World Health Organization; CDC, Disease Control and Prevention; OR, odds ratio.

affecting drinking alcohol. (Table 4). In this study, the sensitivity, specificity, positive predictive value, and negative predictive value were 42.79%, 93.96%, 70.80%, and 82.75.

## Conclusions

According to the results of the present study, the prevalence of alcohol consumption among teenagers was high. Therefore, it is suggested that demographic, family, and psychological factors should be taken into consideration in health programs for the prevention and treatment of alcohol consumption in adolescents.

## Introduction

Adolescents are exposed to many physical and hormonal changes during puberty [1]. They face a range of changes in various physical, psychological, social, and cultural aspects [2]. Many adolescents go through this critical phase safely, but some of them are not able to cope with these conditions and expose themselves to high-risk behaviors such as alcohol consumption [3]. Various factors may affect adolescents' tendency to alcohol, from a behavioral perspective sometimes people drink alcohol to relieve anxiety and loneliness [4–6]. In addition to the turmoil in the family environment, the way family members treat each other and, family history of drug and alcohol consumption are effective factors in this regard [7, 8].

Alcohol consumption is very common in adolescents. A study in Thailand, done on adolescents aged 10–14 evaluated 9509 people and found 30.01% of the adolescents with the experience of alcohol consumption, adolescents whose parents were divorced or neither of their parents lived with them, had a higher chance of alcohol consumption [9]. Another study of 3005 adolescents, 12–17 years old, in Mexico found that 59% of adolescents had experienced alcohol consumption, which was significantly associated with increasing age, low parental supervision, and dropout rates [10]. Also, a study of female students found that almost half of female high school students (12 to 17 years old) had consumed alcohol by then [11]. A study conducted by Getachew et al. [12] on 3967 adolescents in the age group of 13 to 19 in 20 high schools found that 29% of them always consume alcohol. In this study, it was found that one of the factors of a tendency to alcohol in adolescents is to have parents who use tobacco.

A study by Leung et al. [13] examined alcohol consumption and its consequences in 68 low- and middle-income countries. In this study, it was found that alcohol consumption is much more common in men than women. On the other hand, a study by Ferreira et al. [14] on 785 adolescents found that about 25.5% of adolescents had consumed alcohol. Male gender, ages 17 to 19, not living with the mother, using a weight loss strategy in the last 3 months, and especially being a victim of domestic violence were important predictors of alcohol abuse in this study.

Alcohol use in adolescents may be associated with high-risk behaviors [15]. For example, alcohol consumption may lead adolescents to have unprotected sex, which increases the prevalence of unwanted pregnancies and sexually transmitted diseases in adolescents [16]. Also, drinking alcohol in adolescents may lead to high-risk driving and as a result there is an increase in life and financial losses in driving [17]. On the other hand, it can be said that the adolescents who are prone to high-risk behaviors are also more prone to other high-risk behaviors. For example, the adolescents who use alcohol are more likely to use drugs than others in their age group [18] or to have aggressive behaviors, violence and suicidal tendencies

[19]. Therefore, alcohol consumption in these age groups should be taken seriously because it can be the beginning of other social dangers [20].

In addition to the things mentioned above alcohol consumption in adolescence is associated with serious physical harm in adulthood. Alcohol consumption can lead to many physical injuries such as cancer [21]. According to the mentioned points, alcohol consumption should be considered as an important priority of health care organizations as well as an inter-sectoral program in the community. Alcohol consumption as a social issue is very important because it can lead to a variety of violent and anti-social behaviors in adolescents and young people. Hence, the high-risk behavior in adolescents has many destructive effects on both society and the individual, and its use in adolescence can lead to the use of stimulants and traditional drugs in adulthood. Due to the importance of the issue, it is necessary to identify the factors related to alcohol consumption in adolescents in order to develop prevention programs, therefore, the aim of this study was to investigate the prevalence of alcohol consumption and its related factors in adolescents aged 11 to 16 years.

## Methods

In the present study, the Global School-based Student Health Survey(GSHS) data (2003–2018) available for public on the websites of the US Centers for Disease Control and Prevention (CDC) and WHO was used. Methods and the most important findings of the GSHS are explained on both the CDC and WHO websites.

Briefly, the GSHS is a self-administered school-based survey developed by WHO and the CDC. The aim of the GSHS is to provide data on health behaviors among adolescents aged 11–16 years using the same systematized formulae to aid countries in expanding functional health programs and policies. A similar systematized sampling approach, study methodology, and questionnaire were used in all countries.

Briefly, in all countries, attendees were chosen by using a two-stage group design to provide a nationally representative sample of young adolescents aged 11–16 years. All adolescents in the selected school classes were included in the sampling frame. The GSHS questionnaire is made of ten modules of questions on various aspects of health including tobacco use, diet, physical activity, sexual behaviors, and alcohol consumption. Alcohol consumption indicators include (last month's use, ever drunk, two or more drinks in a day, and trouble from drinking).

Countries are free to select different modules when they do a national GSHS. However, when a module is included in a GSHS in a country, all core questions of the module must be used (there are generally less than seven core questions per module) and the wording of the questions cannot be changed (except for translation in the local language). The questionnaire is anonymous and it is self-administered to the adolescents during a 40–45 min period in the classroom. Adolescents record their responses on a computer-scanable answer sheet and data entry is done automatically at the US CDC using an automated optic character recognition procedure.

The GSHS was approved by the Ministry of Education or a Health Research Ethics Committee in each participating country. Participants entered the study voluntarily and verbal or written consent was obtained from all adolescents and their parents or guardians in each country. This study was conducted on data collected from low- and middle-income countries between 2003 and 2018, by World Health Organization guidelines.

### Ethics approval and consent to participate

The study was approved by the ethics committee of the vice chancellery of research and technology, Kermanshah University of Medical Sciences (IR.KUMS.REC.1398.711).

## Data collection and definition of variables

The present study used data from the last accessible GSHS. Lastly, 288385 adolescents were included in this study (Table 1). Variables were classified into six generic categories as follows: socio-demographic factors (such as age, sex, and educational level); mental health factors (loneliness, insomnia due to anxiety or worry, the number of close friends, and suicide plan); protective factors (parental supervision, parental support, parental awareness, peer support, and physical activity); and other background factors (cigarette, times used marijuana, had sex, eating fruit, truancy, bullied, parental used tobacco). The details of the variable description in (Table 1).

## Statistical analysis and calculations

Descriptive statistics were used to report the ratio of each variable in the study population, by gender. Differences in the proportions were compared using chi-square test. Univariate and multivariate logistic regression were used to evaluate the raw and adjusted odds ratio (OR), respectively. Variables with a P value of less than 0.25 in the univariate analysis were entered

**Table 1. Definition of the variables in the study.**

| Variable | Survey questions and coding |
| --- | --- |
| Drinking alcohol | "During the past 30 days, on the days you drank alcohol, how many drinks did you usually drink per day?" |
| Sex | "What is your sex?" |
| Age | "How old are you?" |
| Grade | "In what grade/class/ standard are you?" |
| Truancy | "During the past 30 days, on how many days did you miss classes or school without permission?" |
| Loneliness | "During the past 12 months, how often have you felt lonely?" |
| Insomnia | "During the past 12 months, how often have you been so worried about something that you could not sleep at night?" |
| Daily activity | "During the past 7 days, on how many days were you physically active for a total of at least 60 minutes per day?" |
| Fruit | "During the past 30 days, how many times per day did you usually eat fruit, such as COUNTRY SPECIFIC EXAMPLES?" |
| Bullied | "During the past 30 days, on how many days were you bullied?" |
| Close friend | "How many close friends do you have?" |
| Parent used tobacco | "Which of your parents or guardians use any form of tobacco?" |
| Parental supervision | "During the past 30 days, how often did your parents or guardians check to see if your homework was done?" |
| Parental support | "During the past 30 days, how often did your parents or guardians understand your problems and worries?" |
| Parental awareness | "During the past 30 days, how often did your parents or guardians really know what you were doing with your free time?" |
| Peer support | "During the past 30 days, how often were most of the adolescents in your school kind and helpful?" |
| Cigarette | "During the past 30 days, on how many days did you smoke cigarettes?" |
| Times used marijuana | "During the past 30 days, how many times have you used marijuana (also called COUNTRY SPECIFIC SLANG TERMS FOR MARIJUANA)?" |
| Had sex | "Have you ever had sexual intercourse?" |
| Suicide plan | "During the past 12 months, did you make a plan about how you would attempt suicide?" |

Data source: Global School-based Student Health Survey (2003–2018) reported from WHO.

into multivariate logistic regression. All analyses were performed at a significance level of 5% using Stata software version 14.1 (Stata Corp, College Station, TX, USA)

## Results

### Participants

Out of 371303 students, 288385 answered alcohol questions, the Response Rate was (77.7%). 133182 (46.1. %) of the participants in the study were boys and the rest were girls. The highest response rates were in the Syrian Arab Republic (99.5%), Fiji (99.3%) and Malaysia (99.1%), respectively, and the lowest response rates were in the Seychelles (82.4%), Venezuela (Barinas) (86.9%) and Samoa (87.1%), respectively. The prevalence of alcohol consumption among adolescents dents was 18.2%, which was 25031 (19.2%) in boys and 23570 (17.3%) in girls. Seychelles (57.9%), Colombia Bogot (Official & Privado) (54.8%), and Montserrat (54.2%) had the highest prevalence of alcohol consumption, respectively and Tajikistan (0.7%), Myanmar (1.8%) and Indonesia (National) (2.8%) had the lowest prevalence of alcohol consumption among adolescents (Table 2).

### Prevalence of alcohol consumption

According to Table 2, 23.45% of the male adolescents who consumed alcohol had at least 10 or more days of marijuana use in 30 days, and among (24.74%) of boys and (24.07%) of girls, a suicide plan was observed (Table 3).

### Determinants of alcohol consumption

After adjustment for potential confounders, the odds of drinking alcohol among girls was 34% less than boys (p < .001).

Multivariate analysis showed that the Age for 16 and more than 16 years old (OR = 3.08,95%CI: 2.54–3.74), truancy for more than 10 days (OR = 1.24, 95%CI: 1.08–1.43), loneliness at sometimes of the times (OR = 1.04, 95%CI: 1.01–1.07), Insomnia at most of the times (OR = 1.85, 95%CI: 1.70–2.01), daily activity (OR = 1.03, 95%CI: 1.00–1.07), bullied for 1–9 Days in a month (OR = 1.24, 95%CI: 1.09–1.40), cigarette (OR = 4.01, 95%CI: 3.86–4.17), used marijuana for more than 10 days in a month (OR = 5.58, 95%CI: 4.59–7.78), had sex (OR = 2.76, 95%CI: 2.68–2.84), and suicide plan (OR = 1.48, 95%CI: 1.42–1.54) were important factors affecting drinking alcohol. (Table 4). In this study, the sensitivity, specificity, positive predictive value, and negative predictive value were 42.79%, 93.96%, 70.80%, and 82.75. (Table 4).

## Discussion

Analysis of the findings showed that alcohol consumption is common in adolescents, which was also found in the studies of Benjet et al. [10], Kittipichai et al. [11], and Getachew et al. [12]. In a study of Thai adolescents, Luecha et al. [9] found that 31.01% of adolescents aged 10 to 14 had consumed alcohol at least once. In a study by Ting et al. [22] on 11 to 12-year-old Taiwanese adolescents, alcohol consumption was reported at 48% in this group. Explaining this finding, it can be said that seeking diversity, curiosity, easy access, misconceptions about alcohol, being influenced by friends, and imitating them are among the most important causes of adolescents' tendency to consume alcohol [10–12]. It can also be stated that adolescents are exposed to high-risk behaviors such as alcohol consumption due to special conditions in this age group [23]. Adolescence is a period of changes in physical, sexual, psychological, and

**Table 2. Country-level breakdown of the drinking alcohol prevalence (2003–2013).**

| Country | Year of survey | Gender | Response rate (%) * | Sample size | Prevalence | 95%CI** |
|---|---|---|---|---|---|---|
| Anguilla | 2009 | Boys | 91.7 | 407 | 45.0 | 40–50 |
| | | Girls | 95.2 | 475 | 44.0 | 40–49 |
| | | Total | 93.4 | 888 | 44.7 | 41–48 |
| Argentina National | 2012 | Boys | 91.7 | 12606 | 53.52 | 53–54 |
| | | Girls | 94.9 | 13876 | 53.50 | 53–54 |
| | | Total | 94.6 | 26828 | 53.4 | 53–54 |
| Bahamas | 2013 | Boys | 96.3 | 603 | 27.9 | 24–32 |
| | | Girls | 95.2 | 689 | 26.9 | 24–30 |
| | | Total | 95.7 | 1299 | 27.6 | 25–30 |
| Barbados | 2011 | Boys | 94.6 | 694 | 48.1 | 44–52 |
| | | Girls | 95.6 | 852 | 47.3 | 44–51 |
| | | Total | 95.2 | 1550 | 47.7 | 45–50 |
| Belize | 2011 | Boys | 96.5 | 957 | 32.2 | 29–35 |
| | | Girls | 96.9 | 1076 | 26.7 | 24–29 |
| | | Total | 96.7 | 2042 | 29.3 | 27–31 |
| Benin | 2009 | Boys | 91.8 | 1598 | 24.3 | 22–26 |
| | | Girls | 89.4 | 832 | 16.6 | 14–19 |
| | | Total | 91.0 | 2448 | 21.7 | 20–23 |
| BoliviaNational | 2012 | Boys | 96.4 | 1728 | 20.5 | 19–22 |
| | | Girls | 96.2 | 1676 | 15.6 | 14–17 |
| | | Total | 96.2 | 3557 | 18.3 | 17–20 |
| British Virgin Islands | 2009 | Boys | 95.1 | 701 | 33.8 | 3–37 |
| | | Girls | 96.5 | 886 | 35.9 | 33–39 |
| | | Total | 95.9 | 1596 | 35 | 33–37 |
| Brunei Darus salam | 2014 | Boys | 98.3 | 1189 | 5.7 | 5–7 |
| | | Girls | 98.7 | 1363 | 2.7 | 2–4 |
| | | Total | 98.5 | 2560 | 4.2 | 3–5 |
| Cambodia | 2013 | Boys | 96.9 | 1736 | 17.9 | 16–2 |
| | | Girls | 98.1 | 1965 | 5.1 | 4–6 |
| | | Total | 97.6 | 3713 | 11.1 | 10–12 |
| Cayman Islands | 2007 | Boys | 85.4 | 536 | 40.7 | 37–45 |
| | | Girls | 91.3 | 605 | 37.4 | 34–41 |
| | | Total | 88.3 | 1147 | 39 | 36–42 |
| Chile National | 2013 | Boys | 96.9 | 986 | 32 | 29–35 |
| | | Girls | 96.2 | 976 | 31.7 | 29–35 |
| | | Total | 96.4 | 1976 | 32 | 30–34 |
| China Beijing | 2003 | Boys | 92.0 | 1041 | 18 | 16–2 |
| | | Girls | 94.8 | 1147 | 7.9 | 7–10 |
| | | Total | 93.5 | 2195 | 12.7 | 11–14 |
| Colombia Bogot (Oficial & Privado) | 2007 | Boys | 90.0 | 4023 | 53.2 | 52–55 |
| | | Girls | 92.0 | 4972 | 56.1 | 55–57 |
| | | Total | 91.0 | 9020 | 54.8 | 54–56 |
| Costa Rica | 2009 | Boys | 98.6 | 1269 | 27 | 25–29 |
| | | Girls | 97.9 | 1353 | 25.5 | 23–28 |
| | | Total | 98.2 | 2630 | 26.3 | 25–28 |

*(Continued)*

**Table 2.** (Continued)

| Country | Year of survey | Gender | Response rate (%) * | Sample size | Prevalence | 95%CI** |
|---|---|---|---|---|---|---|
| Dominica | 2009 | Boys | 94.5 | 675 | 54.2 | 5–58 |
| | | Girls | 94.2 | 870 | 49.2 | 46–53 |
| | | Total | 94.3 | 1548 | 51.5 | 49–54 |
| Ecuador Guayaquil | 2007 | Boys | 83.0 | 2196 | 31.9 | 3–34 |
| | | Girls | 84.9 | 2377 | 27.1 | 25–29 |
| | | Total | 83.8 | 4628 | 29.6 | 28–31 |
| El Salvador | 2013 | Boys | 96.1 | 973 | 19.7 | 17–22 |
| | | Girls | 95.9 | 834 | 17.5 | 15–20 |
| | | Total | 95.9 | 1837 | 19 | 17–21 |
| Fiji | 2010 | Boys | 98.9 | 973 | 22.8 | 2–26 |
| | | Girls | 99.6 | 948 | 11.3 | 9–13 |
| | | Total | 99.3 | 1661 | 16.2 | 15–18 |
| Ghana | | Boys | 94.4 | 1005 | 13.3 | 11–16 |
| | | Girls | 93.3 | 847 | 11.8 | 10–14 |
| | | Total | 93.9 | 1863 | 12.6 | 11–14 |
| Guatemala (National) | 2002 | Boys | 95.8 | 2404 | 20.8 | 19–23 |
| | | Girls | 96.9 | 2909 | 16.8 | 15–18 |
| | | Total | 96.3 | 5387 | 18.6 | 18–20 |
| Guyana NATIONAL | 2010 | Boys | 95.0 | 982 | 46.5 | 43–5 |
| | | Girls | 93.8 | 1246 | 36.6 | 34–39 |
| | | Total | 94.3 | 2256 | 41 | 39–43 |
| Honduras | 2012 | Boys | 95.5 | 801 | 14.5 | 12–17 |
| | | Girls | 96.1 | 874 | 16.9 | 15–20 |
| | | Total | 95.7 | 1702 | 16 | 14–18 |
| Indonesia (NATIONAL) | 2007 | Boys | 95.0 | 1405 | 4.9 | 4–6 |
| | | Girls | 98.3 | 1596 | 0.8 | 0.4–1 |
| | | Total | 96.7 | 3014 | 2.8 | 2–3 |
| Jamaica | 2010 | Boys | 94.5 | 735 | 58.5 | 55–62 |
| | | Girls | 94.1 | 769 | 48.8 | 45–52 |
| | | Total | 94.3 | 1531 | 53.5 | 51–56 |
| Kiribati | 2011 | Boys | 94.8 | 651 | 46.9 | 43–51 |
| | | Girls | 96.9 | 861 | 19.9 | 17–23 |
| | | Total | 95.9 | 1517 | 31.5 | 29–34 |
| Lebanon | 2011 | Boys | 97.4 | 1036 | 30.9 | 28–34 |
| | | Girls | 98.1 | 1197 | 18.5 | 16–0.21 |
| | | Total | 97.8 | 2235 | 24.3 | 23–0.26 |
| Malawi (National) | 2009 | Boys | 95.1 | 1002 | 7.8 | 6–10 |
| | | Girls | 95.9 | 1160 | 3.6 | 3–5 |
| | | Total | 95.3 | 2248 | 5.9 | 5–7 |
| Malaysia | 2012 | Boys | 98.8 | 12577 | 9.5 | 9–10 |
| | | Girls | 99.5 | 12665 | 5.7 | 5–6 |
| | | Total | 99.1 | 25285 | 7.6 | 7–8 |
| Maldives (National) | 2009 | Boys | 91.1 | 1323 | 7.9 | 7–9 |
| | | Girls | 94.7 | 1659 | 3.5 | 3–4 |
| | | Total | 92.9 | 2999 | 5.6 | 5–6 |

(*Continued*)

**Table 2.** (Continued)

| Country | Year of survey | Gender | Response rate (%) * | Sample size | Prevalence | 95%CI** |
|---|---|---|---|---|---|---|
| Mauritius (Mauritius) | 2011 | Boys | 98.2 | 972 | 26.1 | 23–29 |
| | | Girls | 98.1 | 1150 | 21.7 | 19–24 |
| | | Total | 98.2 | 2128 | 23.9 | 22–26 |
| Mongolia | 2013 | Boys | 97.9 | 2464 | 9.7 | 9–11 |
| | | Girls | 98.8 | 2820 | 6.5 | 6–7 |
| | | Total | 98.4 | 5306 | 8.1 | 7–9 |
| Montserrat | 2007 | Boys | 88.2 | 754 | 52.5 | 49–56 |
| | | Girls | 92.2 | 1002 | 55.6 | 52–59 |
| | | Total | 90.4 | 1759 | 54.2 | 52–57 |
| Myanmar | 2007 | Boys | 95.8 | 1337 | 3 | 2–4 |
| | | Girls | 98.6 | 1388 | 0.6 | 2–10 |
| | | Total | 97.2 | 2727 | 1.8 | 1–2 |
| Namibia National | 2013 | Boys | 94.1 | 1990 | 38.8 | 37–41 |
| | | Girls | 94.9 | 2237 | 27.5 | 26–29 |
| | | Total | 94.5 | 4284 | 32.8 | 31–34 |
| Nauru | 2011 | Boys | 87.9 | 204 | 27.5 | 22–34 |
| | | Girls | 95.0 | 306 | 26.8 | 22–32 |
| | | Total | 91.9 | 531 | 27.3 | 24–31 |
| Peru | 2010 | Boys | 95.0 | 1332 | 32.4 | 3–35 |
| | | Girls | 94.6 | 1383 | 27.8 | 25–30 |
| | | Total | 94.8 | 2732 | 29.9 | 28–32 |
| Philippines National | 2011 | Boys | 97.2 | 2215 | 27 | 25–29 |
| | | Girls | 98.3 | 2934 | 14.6 | 13–16 |
| | | Total | 97.8 | 5174 | 19.9 | 19–21 |
| Samoa | 2011 | Boys | 86.5 | 839 | 45.3 | 42–49 |
| | | Girls | 88.1 | 1212 | 27.6 | 25–30 |
| | | Total | 87.1 | 2107 | 35.4 | 33–37 |
| Senegal | 2005 | Boys | 94.8 | 1619 | 5.8 | 5–7 |
| | | Girls | 94.5 | 1327 | 1.7 | 1–3 |
| | | Total | 94.6 | 2984 | 3.9 | 3–5 |
| Seychelles | 2007 | Boys | 78.9 | 534 | 59.7 | 56–64 |
| | | Girls | 85.7 | 634 | 56.2 | 52–60 |
| | | Total | 82.4 | 1180 | 57.9 | 55–61 |
| Solomon Islands | 2011 | Boys | 91.3 | 642 | 28.5 | 25–32 |
| | | Girls | 92.8 | 605 | 16.7 | 14–20 |
| | | Total | 91.8 | 1305 | 23.3 | 21–26 |
| Suriname | 2009 | Boys | 90.7 | 784 | 41.1 | 38–45 |
| | | Girls | 87.0 | 720 | 34.3 | 31–38 |
| | | Total | 88.9 | 1510 | 37.7 | 35–40 |
| Syrian Arab Republic | 2010 | Boys | 100.0 | 1243 | 10.5 | 9–12 |
| | | Girls | 99.2 | 1845 | 3.4 | 3–4 |
| | | Total | 99.5 | 3088 | 6.3 | 5–7 |
| Tajikistan | 2006 | Boys | 94.5 | 4542 | 1 | 0.7–1 |
| | | Girls | 96.5 | 4627 | 0.3 | 0.2–0.5 |
| | | Total | 95.4 | 9265 | 0.7 | 0.5–0.9 |

(*Continued*)

**Table 2.** (Continued)

| Country | Year of survey | Gender | Response rate (%) * | Sample size | Prevalence | 95%CI** |
|---|---|---|---|---|---|---|
| Thailand | 2008 | Boys | 86.1 | 1175 | 21 | 19–23 |
| | | Girls | 94.5 | 1319 | 9.2 | 8–11 |
| | | Total | 90.3 | 2498 | 14.8 | 13–16 |
| Trinidad and Tobago (National) | 2011 | Boys | 94.9 | 1442 | 36.3 | 34–39 |
| | | Girls | 94.0 | 1187 | 33.1 | 30–36 |
| | | Total | 94.5 | 2657 | 35 | 33–37 |
| Tuvalu | 2013 | Boys | 92.5 | 420 | 25.2 | 21–30 |
| | | Girls | 95.6 | 461 | 7.4 | 5–10 |
| | | Total | 94.2 | 888 | 15.8 | 14–18 |
| Uganda (National) | 2003 | Boys | 90.2 | 1456 | 16.5 | 15–18 |
| | | Girls | 91.4 | 1395 | 13.3 | 12–15 |
| | | Total | 90.5 | 2910 | 15 | 14–16 |
| Uruguay (NATIONAL) | 2012 | Boys | 94.9 | 1538 | 51.7 | 49–54 |
| | | Girls | 94.6 | 1766 | 47.2 | 45–50 |
| | | Total | 94.7 | 3338 | 49.2 | 47–51 |
| Vanuatu | 2011 | Boys | 95.5 | 464 | 12.5 | 1–16 |
| | | Girls | 97.7 | 603 | 6.6 | 5–9 |
| | | Total | 96.8 | 1083 | 9.3 | 8–11 |
| Venezuela (Barinas) | 2003 | Boys | 83.5 | 886 | 37.1 | 34–4 |
| | | Girls | 90.1 | 1052 | 28.1 | 25–31 |
| | | Total | 86.9 | 1954 | 32 | 30–34 |
| Viet Nam | 2013 | Boys | 95.8 | 1491 | 30.4 | 28–33 |
| | | Girls | 96.2 | 1698 | 15.7 | 14–18 |
| | | Total | 96.0 | 3198 | 22.6 | 21–24 |
| Yugoslav Republic of Macedonia | 2007 | Boys | 87.8 | 908 | 47.2 | 44–51 |
| | | Girls | 90.4 | 953 | 37.7 | 35–41 |
| | | Total | 89.1 | 1884 | 42.2 | 40–44 |
| Zimbabwe | 2003 | Boys | 86.4 | 639 | 23.9 | 21–27 |
| | | Girls | 88.4 | 934 | 16.7 | 14–19 |
| | | Total | 87.5 | 1578 | 19.6 | 18–22 |
| Total | | | 70.9 | 191228 | 18.2 | 18–21 |
| **Country-level breakdown of the drinking alcohol prevalence (2014–2018)** | | | | | | |
| Bangladesh | 2014 | Boys | 98.6 | 1192 | 2 | 2–4 |
| | | Girls | 99.2 | 1788 | 0.4 | 0.2–0.8 |
| | | Total | 98.93 | 2,989 | 1 | 1–2 |
| Benin | 2016 | Boys | 92.8 | 1366 | 45 | 43–49 |
| | | Girls | 93.5 | 1151 | 43 | 40.7–46.7 |
| | | Total | 93.14 | 2,536 | 45 | 43–47 |
| Bhutan | 2016 | Boys | 97.3 | 3384 | 34 | 32–36 |
| | | Girls | 99.2 | 4105 | 17.6 | 16.4–18.7 |
| | | Total | 98.39 | 7,576 | 25 | 24–26 |
| Brunei Darussalam | 2014 | Boys | 98 | 1210 | 5 | 5–7 |
| | | Girls | 98.7 | 1381 | 3.4 | 2.6–4.6 |
| | | Total | 98.38 | 2,599 | 4 | 4–5 |
| Cook Islands | 2015 | Boys | 95.6 | 342 | 36 | 31–42 |
| | | Girls | 93.8 | 354 | 34.3 | 29.4–39.6 |
| | | Total | 94.72 | 701 | 35 | 32–39 |

(*Continued*)

**Table 2.** (Continued)

| Country | Year of survey | Gender | Response rate (%) * | Sample size | Prevalence | 95%CI** |
|---|---|---|---|---|---|---|
| (CUW)USE | 2015 | Boys | 92.6 | 1230 | 35 | 32–38 |
| | | Girls | 91.5 | 1508 | 38 | 35.4–40.6 |
| | | Total | 92.01 | 2,765 | 37 | 35–39 |
| Sierra leone | 2016 | Boys | 91.5 | 636 | 43 | 39–47 |
| | | Girls | 91.2 | 794 | 43.5 | 39.9–47 |
| | | Total | 91.02 | 1,481 | 43 | 41–46 |
| Jamaica | 2017 | Boys | 91.1 | 755 | 55 | 51–59 |
| | | Girls | 94.8 | 900 | 40.7 | 37.4–44 |
| | | Total | 93.16 | 1,667 | 47 | 44–49 |
| Laos | 2015 | Boys | 96.4 | 1668 | 38 | 35–40 |
| | | Girls | 96.4 | 1990 | 33.9 | 32–36.1 |
| | | Total | 96.42 | 3,683 | 36 | 34–37 |
| Lebanon | 2017 | Boys | 93.3 | 2330 | 20 | 19–22 |
| | | Girls | 96.5 | 3370 | 9.6 | 8.7–10.7 |
| | | Total | 95.18 | 5,708 | 14 | 13–15 |
| Liberia | 2017 | Boys | 93.3 | 1382 | 28 | 26–31 |
| | | Girls | 90.3 | 1253 | 25.3 | 22.8–27.9 |
| | | Total | 91.47 | 2,744 | 28 | 26–30 |
| Mauritania national | 2017 | Boys | 95.8 | 1414 | 24 | 22–26 |
| | | Girls | 96.4 | 1584 | 26 | 23.9–28.3 |
| | | Total | 96.12 | 3,012 | 25 | 24–27 |
| Mozambique | 2015 | Boys | 94.9 | 994 | 16 | 14–19 |
| | | Girls | 95.2 | 870 | 12.6 | 10.5–15 |
| | | Total | 95.05 | 1,918 | 14 | 13–16 |
| Myanmar | 2016 | Boys | 98.3 | 1301 | 10 | 9–12 |
| | | Girls | 99.6 | 1511 | 3.1 | 0.2–4.1 |
| | | Total | 98.94 | 2,838 | 7 | 6–8 |
| Nepal | 2015 | Boys | 98.5 | 3016 | 7.5 | 6.6–8.5 |
| | | Girls | 97.8 | 3406 | 3.9 | 3.3–4.6 |
| | | Total | 98.1 | 6529 | 5.7 | 5.2–6.3 |
| Jamaica | 2017 | Boys | 93.4 | 1468 | 38 | 36–41 |
| | | Girls | 93.5 | 1619 | 36.8 | 34.4–39.3 |
| | | Total | 93.3 | 3149 | 37.3 | 36–39 |
| Philippines | 2015 | Boys | 96.8 | 3991 | 29 | 28–31 |
| | | Girls | 97.6 | 4769 | 19.7 | 18.5–20.8 |
| | | Total | 97.21 | 8,761 | 24 | 23–25 |
| Saint Luzia | 2018 | Boys | 91.5 | 909 | 45 | 42–49 |
| | | Girls | 93.9 | 1044 | 44.9 | 42–48 |
| | | Total | 92.79 | 1,970 | 0.45 | 0.43–0.48 |
| Seychelles | 2015 | Boys | 86.8 | 2404 | 44 | 43–47 |
| | | Girls | 91.8 | 2674 | 4.7 | 45.3–49.3 |
| | | Total | 89.45 | 5,080 | 46 | 45–48 |
| Sierra leone | 2017 | Boys | 94.3 | 1258 | 20 | 18–23 |
| | | Girls | 94.3 | 1484 | 12.9 | 11.2–14.7 |
| | | Total | 94.28 | 2,798 | 17 | 15–18 |

(*Continued*)

**Table 2.** (Continued)

| Country | Year of survey | Gender | Response rate (%) * | Sample size | Prevalence | 95%CI** |
|---|---|---|---|---|---|---|
| Sri Lanka | 2016 | Boys | 98.1 | 1437 | 6 | 5.1–7.7 |
| | | Girls | 99.4 | 1805 | 1.2 | 0.8–1.9 |
| | | Total | 98.74 | 3,262 | 4 | 3–4 |
| Suriname | 2016 | Boys | 93.6 | 1040 | 42 | 39–46 |
| | | Girls | 94.5 | 1072 | 38.2 | 35.3–41.2 |
| | | Total | 93.89 | 2,126 | 40 | 38–42 |
| Timor- leste | 2015 | Boys | 93.8 | 1625 | 30 | 29–33 |
| | | Girls | 94.1 | 1877 | 15.6 | 14–17.3 |
| | | Total | 93.57 | 3,704 | 23 | 22–24 |
| Toxelau | 2014 | Boys | 93.8 | 65 | 44 | 32–57 |
| | | Girls | 95.7 | 70 | 46.3 | 34.8–58.2 |
| | | Total | 95 | 140 | 46 | 38–54 |
| Tonga | 2017 | Boys | 96.7 | 1520 | 22 | 20–24 |
| | | Girls | 98.2 | 1792 | 9.3 | 8.1–10.8 |
| | | Total | 97.48 | 3,333 | 15 | 14–16 |
| Trinidad & Tobago | 2017 | Boys | 92.3 | 1790 | 30 | 28–32 |
| | | Girls | 95.3 | 2050 | 30.4 | 28.4–32.5 |
| | | Total | 93.9 | 3,869 | 30 | 29–32 |
| Tanzania | 2014 | Boys | 97 | 1782 | 7 | 6–9 |
| | | Girls | 97 | 1935 | 6.7 | 5.6–7.9 |
| | | Total | 96.76 | 3,793 | 7 | 6–8 |
| Wallis & Futuna Islands | 2015 | Boys | 91 | 531 | 40 | 36–44 |
| | | Girls | 92.1 | 572 | 35 | 30.6–38.7 |
| | | Total | 91.5 | 1,117 | 37 | 34–40 |
| Samoa | 2017 | Boys | 92.6 | 707 | 16 | 14–20 |
| | | Girls | 96 | 1197 | 9.1 | 7.5–10.9 |
| | | Total | 94.5 | 1,955 | 12 | 10–13 |
| Total | | | 95.6 | 97157 | 22.2 | 22.0–22.5 |

Data source: Global School-based Student Health Survey (2003–2018) reported from WHO.

*Response rate (%): is defined as the percentage of the eligible sampled adolescents of the survey population who responded to this survey.

**95% (CI) = 95% Confidence Intervals.

cognitive development, as well as changes in social needs, lack of appropriate conditions for passing this critical stage can lead to a tendency to consume alcohol [24].

Another finding of the present study was that the prevalence of alcohol consumption in boys is higher than in girls, this finding was consistent with the findings of Assanangkornchai et al. [25], Chaveepojnkamjorn et al. [26], Georgie et al. [27], Pengpid & Peltzer [28]. Explaining this finding, we can point to the biological differences in alcohol consumption between men and women. Compared to men, women generally have less water in their bodies, which is why women reach the peak with less consumption, even if they consume the same amount as men, and this causes men to consume more [29, 30]. We can also point to cultural differences, because in most societies drinking alcohol is masculine, and some men are better accepted by drinking alcohol in the company of their friends and have stronger personal relationships [31, 32]. Social control is greater for women, and women are concerned that alcohol consumption may affect their family relationships and general behavior or make them sexually vulnerable [33, 34].

**Table 3. Characteristics of adolescents aged 11 to 16 years by sex.**

| Variables | Categories | Boys | | Girls | |
|---|---|---|---|---|---|
| | | Alcohol N (%) | Nonalcohol, N (%) | Alcohol, N (%) | Nonalcohol N (%) |
| Age | ≤ 11 years old | 323 (27.68) | 1167 (72.32)* | 270 (14.93) | 1539 (85.07)* |
| | 12 years old | 1256 (20.35) | 6173 (79.65) * | 1193 (12.54) | 8319 (87.46)* |
| | 13 years old | 4265 (22.17) | 19238 (77.83) * | 4615 (16.69) | 23034 (83.31)* |
| | 14 years old | 7344 (31.58) | 23258 (68.42) * | 7775 (21.68) | 28095 (78.32)* |
| | 15 years old | 8667 (41.35) | 20960 (58.65) * | 8624 (25.64) | 25014 (74.36)* |
| | ≥ 16 years old | 13764 (54.38) | 25309 (45.62) * | 10512 (26.26) | 29525 (73.74)* |
| Grade | Grade 1 | 6875 (22.48) | 23705 (77.52) ** | 6305 (18.88) | 27093 (81.12)** |
| | Grade 2 | 9116 (22.48) | 25435 (73.62) ** | 8783 (22.55) | 30158 (77.45) ** |
| | Grade 3 | 9627 (22.48) | 22608 (70.13) ** | 9479 (25.76) | 27321 (74.24) ** |
| | Grade 4 | 5567 (22.48) | 12976 (69.98) ** | 4913 (22.89) | 16547 (77.11) ** |
| | Grade = >5 | 4142 (22.48) | 10751 (72.19) ** | 3169 (18.94) | 13567 (81.06) ** |
| Truancy | No | 16938 (22.11) | 59661 (77.89)* | 16954 (18.12) | 76634 (81.88) * |
| | 1–5 Days | 10663 (35.45) | 19415 (64.55) * | 8321 (29.36) | 20018 (70.64) * |
| | ≥ 6 days | 2397 (50.23) | 2375 (49.77) * | 1855 (2111) | 2111 (53.23) * |
| Loneliness | Never | 11267 (22.54) | 38724 (77.46)* | 6359 (22.54) | 36436 (85.14)* |
| | Rarely or sometimes | 17010 (22.54) | 44545 (72.37) * | 16982 (22.54) | 60532 (78.09)* |
| | Mostly or always | 3775 (22.54) | 7168 (65.50) * | 5426 (22.54) | 12389 (69.54)* |
| Insomnia | Never | 10981 (20.81) | 41789 (79.19)* | 6109 (40633) | 40633 (86.93) * |
| | Rarely or sometimes | 17225 (29.01) | 4215- (70.99) * | 17698 (23.28) | 58309 (76.72) * |
| | Mostly or always | 3594 (38.19) | 5818 (61.81) * | 4903 (33.99) | 9523 (66.01) * |
| Daily activity | No | 22368 (27.18) | 59918 (72.82)* | 22772 (27.77) | 78977 (77.62)* |
| | Yes | 11913 (27.97) | 29097 (70.95) | 9116 (27.26) | 29026 (76.10) |
| Fruit | No | 24806 (28.13) | 63371 (71.87) | 22834 (22.61) | 22834 (22.61) |
| | Yes | 10299 (27.93) | 26570 (72.07) | 9749 (24.21) | 30524 (75.79) |
| Bullied | No | 20818 (25.25) | 61638 (74.75)* | 19846 (25.25) | 77527 (79.62) * |
| | 1–9 Days | 9931 (25.25) | 20762 (67.64) * | 9018 (25.25) | 23160 (71.97) * |
| | ≥ 10 days | 1897 (25.25) | 3200 (62.78) * | 1763 (25.25) | 3240 (64.76) * |
| Close friend | No | 11088 (25.27) | 32279 (74.43) | 9827 (19.16) | 41464 (80.84) |
| | Yes | 20722 (26.38) | 57817 (73.62) | 18845 (21.72) | 67908 (78.28) |
| Parent used tobacco | Neither | 17500 (23.59) | 56674 (76.41)* | 59028 (23.59) | 66080 (52.82) * |
| | Father or Mother or Both | 11753 (23.59) | 24916 (67.95) * | 31559 (23.59) | 32207 (50.51) * |
| | Do not know | 2938 (23.59) | 6584 (69.15) * | 7069 (23.59) | 7392 (51.12) * |
| Parental supervision | Never | 8651 (30.37) | 19836 (69.63)** | 9069 (30.37) | 24363 (72.87) ** |
| | Rarely or sometimes | 12407 (30.37) | 29877 (70.66) ** | 10458 (30.37) | 36373 (77.67) ** |
| | Mostly or always | 8929 (30.37) | 31615 (77.98) ** | 7607 (30.37) | 37776 (83.24) ** |
| Parental support | Never | 6826 (26.95) | 18505 (73.05) | 6478 (78.95) | 20060 (75.59) |
| | Rarely or sometimes | 12690 (690.95) | 30586 (70.68) | 11302 (302.95) | 37614 (76.90) |
| | Mostly or always | 10422 (422.95) | 32106 (75.49) | 9245 (45.95) | 40722 (81.50) |
| Parental awareness | Never | 6869 (29.81) | 16170 (70.19)** | 5795 (29.81) | 16627 (74.15)** |
| | Rarely or sometimes | 12547 (29.81) | 29514 (70.17) ** | 10843 (29.81) | 35120 (76.41)** |
| | Mostly or always | 10449 (29.81) | 35464 (77.24) ** | 10343 (29.81) | 46564 (81.82)** |
| Peer support | Never | 3977 (27.12) | 10685 (72.88) | 2630 (27.12) | 8928 (77.25) |
| | Rarely or sometimes | 14641 (27.12) | 38045 (72.21) | 13369 (27.12) | 43902 (76.66) |
| | Mostly or always | 11288 (27.12) | 32478 (74.21) | 11032 (27.12) | 45700 (80.55) |
| Cigarette | Never Smoked | 14485 (17.18) | 69848 (82.82)* | 17644 (17.18) | 95125 (84.35)* |
| | Smoked | 16220 (17.18) | 14772 (47.66)* | 11184 (17.18) | 7600 (40.46)* |

*(Continued)*

**Table 3.** (Continued)

| Variables | Categories | Boys | | Girls | |
|-----------|-----------|------|------|-------|------|
| | | Alcohol N (%) | Nonalcohol, N (%) | Alcohol, N (%) | Nonalcohol N (%) |
| Used marijuana | Neither | 18577 (23.45) | 60564 (76.55)* | 18577 (23.45) | 74,234 (79.90)* |
| | 3–9 times | 2577 (23.45) | 829 (27.14) * | 1577 (23.45) | 421 (25.09)* |
| | ≥ 10 days | 1577 (23.45) | 463 (18.94) * | 1577 (23.45) | 229 (18.20)* |
| Had sex | No | 12093 (17.68) | 56300 (82.32)* | 12408 (17.68) | 76451 (86.04)* |
| | Yes | 15110 (17.68) | 15155 (50.07)* | 15101 (17.68) | 10310 (40.57)* |
| Suicide plan | No | 25189 (24.18) | 78980 (75.82)* | 20258 (24.25) | 92824 (82.09)* |
| | Yes | 5174 (24.74) | 8439 (61.99)* | 7507 (24.07) | 12687 (62.83)* |

Data source: Global School-based Student Health Survey (2003–2018) reported from WHO.

Chi-square test for equality of proportions p-value reported.

* $P<0.001$

** $P< 0.05$

The analysis of the findings also showed that the highest prevalence was in Seychelles with 57.9%. This finding was consistent with the research done by Perdrix et al. [35] In this study, the prevalence of alcohol consumption in Seychelles was 51.1%. In a study by Pengpid et al., [36] the prevalence of alcohol in adolescents in Seychelles was estimated at 47.6%. The study by Ma et al., [37] which surveyed 13-15-year-old adolescents, found that the prevalence of alcohol in Seychelles was 61.1%. Cultural factors can be mentioned in explaining this finding. Research by Pedrix et al. [35] states that alcohol consumption in Seychelles is common at many parties and family and social events, and the availability of alcohol affects their desire to consume alcohol. It can also be said that since Seychelles economy is based on tourism, the cultural acceptance of other countries may have been effective in this prevalence.

On the other hand, the lowest prevalence was found in Tajikistan with 0.7%. In explaining this finding, the role of religious factors can be mentioned. In Tajikistan, Islam is the predominant religion, and in Islamic countries, alcohol consumption is prohibited and the buying and selling of alcohol is legally punishable. In Iran, which is an Islamic country, a study conducted by AMIN-Amin-Esmaeili et al. [38] estimated the prevalence of alcohol over the past year and the past week at 5.7% and 1%, respectively. In Islamic countries, alcohol consumption is considered a sin religion is a deterrent to alcohol consumption and even alcohol advertising is a violation [39, 40]. Therefore, according to religious and cultural beliefs about alcohol consumption in Islamic countries, adolescents living in these countries may be less inclined to consume alcohol.

The study also found that marijuana use in adolescents greatly increases the chances of alcohol consumption. In a study conducted by Sokolovsky et al., [41] 341 young university students were surveyed. In this study, it was found that marijuana and alcohol are often used simultaneously and their simultaneous use has more negative consequences. There is a two-way relationship between alcohol consumption and marijuana use, adolescents who use marijuana may also use more alcohol, and vice versa, usually marijuana and alcohol are consumed simultaneously. In general, it can be said that performing a high-risk behavior in adolescence can lead to different behaviors [42–44].

Another finding of the present study was that smoking increases the likelihood of alcohol consumption. The same finding was found in the studies of Thrul et al. [45], McKee et al. [46], Piasecki et al. [47]. In this regard, Thrul et al. [45] in a study showed that the simultaneous consumption of cigarettes and alcohol increases the perception of rewards for consumption

**Table 4. Univariate and multivariate logistic regression analysis of drinking alcohol.**

| Variables | | Crude OR (95%CI) * | Adjusted OR (95%CI) * | P |
|---|---|---|---|---|
| Sex (ref = boy) | Boy | 1 | 1 | |
| | Girl | 0.76 (0.75–0.77) | 1.6 (1.03–1.09) | <0.001 |
| Age(ref: ≤ 11 years old) | 12 years old | 0.66 (0.61–0.72) | 0.87 (0.71–1.06) | 0.172 |
| | 13 years old | 0.88 (0.81–0.95) | 1.23 (1.02–1.50) | 0.032 |
| | 14 years old | 1.27 (1.17–1.37) | 1.96 (1.62–2.38) | <0.001 |
| | 15 years old | 1.61 (1.49–1.74) | 2.74 (2.26–3.32) | <0.001 |
| | ≥ 16 years old | 1.86 (1.72–2.01) | 3.08 (2.54–3.74) | |
| Grade | Grade 1 | 1 | 1 | |
| | Grade 2 | 1.22 (1.20–1.25) | 0.88 (0.84–0.92) | <0.001 |
| | Grade 3 | 1.37 (1.34–1.40) | 0.74 (0.71–0.77) | <0.001 |
| | Grade 4 | 1.14 (1.11–1.17) | 0.35 (0.33–0.37) | <0.001 |
| | Grade 5 | 1.08 (1.05–1.11) | 0.33 (0.31–0.35) | <0.001 |
| Truancy | No | 1 | 1 | |
| | 1–9 Days | 1.59 (1.56–1.61) | 1.24 (1.20–1.28) | <0.001 |
| | ≥ 10 days | 2.22 (2.05–2.39) | 1.24 (1.08–1.43) | 0.003 |
| Loneliness | Never | 1 | 1 | |
| | Rarely or sometimes | 1.34 (1.32–1.37) | 1.04 (1.01–1.07) | 0.018 |
| | Mostly or always | 1.49 (1.43–1.55) | 1.11 (1.03–1.21) | .007 |
| Insomnia | Never | 1 | 1 | |
| | Rarely or sometimes | 1.69 (1.66–1.71) | 1.45 (1.41–1.50) | <0.001 |
| | Mostly or always | 1.95 (1.86–2.04) | 1.85 (1.70–2.01) | <0.001 |
| Daily activity | No | 1 | 1 | |
| | Yes | 1.02(1.00–1.04) | 1.03 (1.00–1.07) | 0.032 |
| fruit | No | 1 | 1 | |
| | Yes | 1.08 (1.06–1.10) | 0.94 (0.92–0.97) | <0.001 |
| Bullied | No | 1 | 1 | |
| | 1–9 Days | 1.37 (1.35–1.40) | 1.07 (1.04–1.10) | <0.001 |
| | ≥ 10 days | 1.42 (1.33–1.52) | 1.24 (1.09–1.40) | 0.001 |
| Close friend | No | 1 | | |
| | Yes | 0.90 (.87-.93) | 1.34 (1.26–1.43) | 0.001 |
| Parent used tobacco | Neither | 1 | 1 | |
| | Father or Mother or Both | 1.37 (1.34–1.39) | 1.07 (1.04–1.10) | <0.001 |
| | Do not know | 3.18 (2.05–4.93) | 1.24 (1.09–1.40) | <0.001 |
| Parental supervision | Never | 1 | 1 | |
| | Rarely or sometimes | .79 (.77-.80) | 0.79 (0.76–0.81) | <0.001 |
| | Mostly or always | .40 (.38-.41) | 0.60 (0.57–0.64) | <0.001 |
| Parental support | Never | 1 | | |
| | Rarely or sometimes | 1.06 (1.04–1.08) | 1.11 (1.07–1.15) | <0.001 |
| | Mostly or always | .53 (.51-.55) | 0.98 (0.92–1.04) | 0.483 |
| Parental awareness | Never | 1 | 1 | |
| | Rarely or sometimes | .89 (.87-.91) | 1.04 (1.00–1.08) | .077 |
| | Mostly or always | .46 (.45-.48) | 1.03 (0.97–1.09) | .396 |
| Peer support | Never | 1 | 1 | |
| | Rarely or sometimes | 1.07 (1.04–1.09) | 1.35 (1.29–1.41) | <0.001 |
| | Mostly or always | .62 (.60-.65) | 0.92 (0.86–0.98) | 0.009 |
| cigarette | No | 1 | 1 | |
| | Yes | 7.41(7.29–7.58) | 4.01 (3.86–4.17) | <0.001 |

*(Continued)*

**Table 4.** (Continued)

| Variables | | Crude OR (95%CI) * | Adjusted OR (95%CI) * | P |
|---|---|---|---|---|
| Used marijuana | Neither | 1 | 1 | |
| | 3–9 times | 5.89 (5.72–6.07) | 5.58 (4.59–6.78) | <0.001 |
| | ≥ 10 days | 4.96 (4.54–5.42) | 4.10 (3.78–4.45) | <0.001 |
| Had sex | No | 1 | 1 | |
| | Yes | 4.15(4.07–4.24) | 2.76 (2.68–2.84) | <0.001 |
| Suicide plan | No | 1 | 1 | |
| | Yes | 2.05 (2.01–2.09) | 1.48 (1.42–1.54) | <0.001 |

Sensitivity: 42.79%, Specificity: 93.96%

Positive predictive value: 70.80%, Negative predictive value:82.75%

Correctly classified:80.90%

95% (CI) * = 95 Confidence Intervals. The variables with a P-value of < 0.25 in the univariate analysis were introduced into the multivariate analysis.

[45]. On the other hand, some studies have shown that the simultaneous consumption of alcohol and cigarettes may have a greater effect on the mesolimbic system, which in turn stimulates the reward system in the brain, and further stimulation of this system leads to increased adolescent desire to consume these two simultaneously [48, 49].

Having sex was another factor that increased the chances of consuming alcohol, alcohol consumption can be associated with engaging in sexual experiences [50]. In a study, Dogan et al. [51] showed that alcohol consumption in adolescents affects the number of sexual partners. Explaining this finding, it can be said that having a positive attitude towards a behavior affects the likelihood of doing that behavior. In some adolescents, there is a view that alcohol consumption has a positive effect on sexual experiences and this view leads to alcohol consumption among them [52]. Some people also believe that alcohol consumption causes a pleasurable sexual relationship and increases sexual attraction and the positive aspects of sexual behavior, all of these factors affect the increase in alcohol consumption [44, 53].

Being alone also increased alcohol consumption. Consistent with this finding, McKay et al. [54], in a study showed that being alone is effective in alcohol consumption. In addition, loneliness and gender are associated with alcohol consumption, so being a woman and experiencing loneliness puts a person at greater risk for drinking alcohol. For example, several studies have shown that alone adolescents may use alcohol, cigarettes, and illegal drugs, probably Adolescents use alcohol as a form of self-medication to reduce loneliness [55, 56].

Another factor that was shown to be effective in alcohol consumption was insomnia. This finding is consistent with Barrow's research [56, 57] which showed that insomnia creates a vicious circle with alcohol consumption. As insomnia increases the risk of alcohol consumption, alcohol consumption can become problematic as well, which exacerbates insomnia. The same finding was found in the research of Roehrs et al. [58] Their study found that people who experience insomnia use alcohol to improve sleep quality. On the other hand, it has been found that a history of committing suicide increases the chances of alcohol consumption. This finding is consistent with the research of Pompili et al. [59] who showed in their research that there is a two-way relationship between alcohol consumption and suicide. Explaining this finding, it can be stated that according to studies, suicide is directly related to anxiety and depression, and many people with a history of suicide use alcohol as self-medication, and when they suffer from anxiety, low mood, or life problems, they turn to alcohol to forget their problems. However, constant use of alcohol can cause tolerance, dependence, and ultimately addiction in the individual [60]. Although alcohol consumption can temporarily reduce

suicidal ideation, in fact, it makes the problem worse. In most cases, long-term alcohol abuse makes suicidal ideation more frequent and powerful and increases the likelihood of attempting suicide [61, 62]. In addition, alcohol abuse generally exacerbates the other factors influencing suicide. For example, alcohol exacerbates the symptoms of many disorders, such as bipolar disorder, borderline personality disorder, and depression, all of which can contribute to suicide. Alcoholism can also cause problems at work, within the family, interpersonal relationships, and the legal system, these problems affect suicide [61].

On the other hand, the effect of daily activity on alcohol consumption cannot be ignored. In this regard, in a study, Conroy et al. [63] showed that after controlling age and gender, daily physical activity was associated with alcohol consumption. If a curious and energetic teenager does not have good entertainment and it is not possible for him/her to have proper daily activities, he/she will be drawn to activities and entertainment that are not good. Therefore, addressing the issue of alcohol consumption in adolescents and young people and preventing it by emphasizing the role of daily activities is very necessary and important.

It has also been found that a history of being beaten increases the chances of alcohol consumption in adolescents. Studying the research related to the long-term effects of child abuse has shown that most adolescents and adults who have had traumatic events as children are more likely to consume alcohol than others. Research shows that childhood abuse experiences can have long-term effects on all aspects of health, development, and well-being [64], and can lead to impaired performance and high-risk behaviors such as alcohol consumption [65]. Waner et al. [66] believe that prolonged exposure to bullying predisposes the child to violence and high-risk behaviors in the future. Children who have been abused have also been found to be more aggressive and delinquent than their peers. These children are pessimistic about their social networks. These signs may be influential in shaping the tendency to consume alcohol [67].

It was also found that parental supervision is a deterrent and effective factor in reducing alcohol consumption in adolescents. The same finding was obtained in the study of Benjet et al. [10]. A study by Strunin et al. [68] also found that parental supervision was effective in limiting alcohol consumption in adolescents. Parental supervision has a significant effect on delaying the tendency to consume alcohol. Adolescents and young people who have less family support and supervision show self-destructive behaviors. The higher the level of family support in adolescents is, the less they are exposed to alcohol. In fact, parental supervision is a protective factor against alcohol consumption [69, 70].

## Strengths and limitations

This study has several strengths, including the fact that a standard questionnaire was used to measure the prevalence of alcohol among adolescents and the samples were selected by a scientific method. Furthermore, the sample size was high and various risk factors for alcohol use were examined, also, because the countries included in the study were culturally, religiously, socially, demographically, and healthily diverse, these cases led to the study of various factors that affected the prevalence of alcohol in adolescents. In addition to the cases mentioned, this study also faced other limitations. First, a self-report questionnaire was used to measure the prevalence of alcohol in adolescents, which may have led to bias in the answers, because the prevalence of alcohol may have been underestimated and adolescents may have concealed their alcohol use or not mentioned the factors affecting it for fear of being reprimanded by their parents and school teachers. Second, because the GSHS did not provide information on the prevalence of alcohol in the parents of these students, this important influencing factor has not been investigated. Third, in this study, it was found that marijuana use, smoking, and having sex are some of the main factors affecting the prevalence of alcohol in adolescents, but

more information in this regard, such as the age of first smoking, marijuana use or sexual intercourse and how often they were done were not available, finally, for the purpose of this study, the missing data were not replaced by statistical methods and therefore such data were removed from the analysis. The results of this study and previous reports showed that using tobacco and smoking cigarettes are affected by various factors. Therefore, to prevent the spread of alcohol in adolescents, various factors and their risk should be considered as well, all these factors should be considered in the development of treatment and prevention programs. It is also suggested that policymakers and therapists of children and adolescents pay special attention to the prevalence of alcohol in adolescence and the factors affecting it, particularly in this study it was found that using marijuana increases the risk of alcohol consumption, which is important and should be considered for prevention. Also, more research is needed to provide better interventions to reduce alcohol consumption in adolescents.

## Conclusion

Due to the importance of the prevalence of alcohol in adolescents and its role in creating high-risk behavior in adulthood, identifying and controlling the factors associated with it is of great importance. Using marijuana, having sex, loneliness, insomnia, suicide plans, and being beaten were among the most important factors associated with adolescent alcohol use. Therefore, according to the factors that have been found to have a greater impact on the prevalence of alcohol consumption, it is recommended to policymakers in this field to design strategic plans to prevent the tendency to drink alcohol in adolescents and to implement part of it in educational environments such as schools. Among these programs, it can be mentioned to inform adolescents about the dangers and harms of consuming alcohol, and marijuana, increasing personal and social skills to reduce loneliness. It is also suggested to the therapists and counselors who are active in the field of adolescents, to reduce alcohol consumption, based on the individual and family factors mentioned in this article, to develop treatment protocols such as improving life skills in adolescents or preparing educational books to increase knowledge of parents.

## Acknowledgments

We sincerely thank the World Health Organization and US Centers for Disease Control for making the GSHS dataset available for free download on their website.

## Author Contributions

**Conceptualization:** Vahid Farnia, Touraj Ahmadi Jouybari, Mehdi Moradinazar.

**Formal analysis:** Fatemeh Khosravi Shadmani, Bahareh Rahami, Tahereh Mohammadi Majd.

**Methodology:** Mehdi Moradinazar, Tahereh Mohammadi Majd.

**Project administration:** Mostafa Alikhani.

**Software:** Tahereh Mohammadi Majd.

**Supervision:** Vahid Farnia.

**Writing – original draft:** Safora Salemi.

**Writing – review & editing:** Vahid Farnia, Safora Salemi, Shahab Bahadorinia.

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
