## [Decision Letter · Decision Letter 0]

10 Apr 2023

PONE-D-23-03514The prevalence of alcohol consumption and its related factors in adolescents: Data extraction from 55 countriesPLOS ONE

Dear Dr. salemi,

Thank you for submitting your manuscript to PLOS ONE. After careful consideration, we feel that it has merit but does not fully meet PLOS ONE’s publication criteria as it currently stands. Therefore, we invite you to submit a revised version of the manuscript that addresses the points raised during the review process. While revising the manuscript based on the comments, please ignore comment no. 6&7 by reviewer 1 which don't look relevant. But try to address all other comments from reviewer 1. For this manuscript to be considered further, including more recent data as commented by reviewer 2 seems to be warranted, please consider that as well as all other comments from reviewer 2.The conclusion in the abstract is very generic, please adjust as per the conclusion in the main body of the manuscript. In addition, once you have considered the comments from one of the reviewers to reanalyze the data, please revisit the key findings of the analysis after and then try to draw specific conclusion and recommendation for the manuscript. If possible try to draw categorical recommendations for policy makers and program managers/implementers specifically unlike a general recommendation now.

We look forward to receiving your revised manuscript.

Kind regards,

Krishna Kumar Aryal

Academic Editor

PLOS ONE

Additional Editor Comments:

In addition, to the following comments by the reviewers, please consider this. The conclusion in the abstract is very generic, please adjust as per the conclusion in the main body of the manuscript. In addition, once you have considered the comments from one of the reviewers to reanalyze the data, please revisit the key findings of the analysis after and then try to draw specific conclusion and recommendation for the manuscript. If possible try to draw categorical recommendations for policy makers and program managers/implementers specifically unlike a general recommendation now.

Reviewers' comments:

Reviewer's Responses to Questions

**Comments to the Author**

1. Is the manuscript technically sound, and do the data support the conclusions?

Reviewer #1: Yes

Reviewer #2: Partly

2. Has the statistical analysis been performed appropriately and rigorously? 

Reviewer #1: Yes

Reviewer #2: Yes

3. Have the authors made all data underlying the findings in their manuscript fully available?

Reviewer #1: Yes

Reviewer #2: Yes

4. Is the manuscript presented in an intelligible fashion and written in standard English?

Reviewer #1: No

Reviewer #2: Yes

5. Review Comments to the Author

Reviewer #1: In this study, the authors have analyzed GSHS data from 55 countries and reported the risk factors leading to alcohol consumption. Followings are my inputs.

1- The title of the MS is not clear. I would suggest revise this like "The prevalence of alcohol consumption and its related factors in adolescents: findings from Global School Health Surveys of 55 countries".

2- The authors should have used a uniform term for study participants either students or adolescents.

3- In abstract under heading methods, it is written that "this cross-section study was performed on ..... is not correct.

4- Similarly in ethic statement, it is written that "written informed consent was obtained from each

participant" which is not correct. Authors have not directly taken written consent, therefore, delete this.

5- The selection criteria of 55 countries is not mentioned. %

6- In routine rate response rate is given for the survey, not for a few questions and a rate above 80% is considered acceptable. Therefore, the term high and low response rate are confusing.

7- The table 2 is redundant and may be deleted.

8- In table 3, the analysis has been given as alcohol and nonalcohol. However, there is no description available in methods.

Reviewer #2: • Authors have considered the data from 2003 to 2013 in the analysis. Authors have not provided any reasons for not including data from countries where the survey was completed after 2013. The latest dataset used in the study dates back approximately 9 years from now, and the prevalence could have substantially changed in the period. I suggest authors to include latest datasets and reanalyze it to make the findings more relevant to present context. In doing so, authors may choose to drop some of the oldest datasets from the analysis process to make sure that data are comparable across countries.

• Authors have pooled data from countries where the survey was done upto 10 years apart. So, comparison of prevalence of tobacco use across countries looks less logical.

• Also, it would be good to clearly state the percentage of variation explained by multivariable regression model.

6. PLOS authors have the option to publish the peer review history of their article (what does this mean?). If published, this will include your full peer review and any attached files.

Reviewer #1: No

Reviewer #2: No

---

## [Author Response · Author response to Decision Letter 0]

29 May 2023

We appreciate you and the reviewers for your precious time in reviewing our paper and providing valuable comments. It was your valuable and insightful comments that led to possible improvements in the current version. The authors have carefully considered the comments and tried our best to address every one of them.

---

## [Editor Report · Decision Letter 1]

31 May 2023

PONE-D-23-03514R1The prevalence of alcohol consumption and its related factors in adolescents: findings from Global School Health Surveys (GSHS) of 55 countriesPLOS ONE

Dear Dr. salemi,

Thank you for submitting your manuscript to PLOS ONE. After careful consideration, we feel that it has merit but does not fully meet PLOS ONE’s publication criteria as it currently stands. Therefore, we invite you to submit a revised version of the manuscript that addresses the points raised during the review process.

We look forward to receiving your revised manuscript.

Kind regards,

Krishna Kumar Aryal

Academic Editor

PLOS ONE

Additional Editor Comments:

Thank you for addressing majority of the comments by the reviewers and editor. However, one of the comments about why the authors have only considered the dataset until 2013 does not look to be addressed. The authors have explained the rationale of including data from long back in the past, but there was not satisfactory explanation or revision of non-inclusion of data after 2013. It just says that during the investigations that the researchers conducted. But the datasets being readily available in the CDC websites and since its already 2023 and running the analysis with a few additional datasets should not be a big issue if the authors have codes for data management and analysis. Please consider that comment from the reviewer 2 which seems to be inadequately addressed. Kindly revisit this part.

---

## [Author Response · Author response to Decision Letter 1]

17 Jul 2023

Dear reviewers

The authors have carefully considered the comments and tried our best to address every one of them.

With respect

---

## [Editor Report · Decision Letter 2]

18 Jul 2023

PONE-D-23-03514R2The prevalence of alcohol consumption and its related factors in adolescents: findings from Global School Health Surveys (GSHS) of 55 countriesPLOS ONE

Dear Dr. salemi,

Thank you for submitting your manuscript to PLOS ONE. After careful consideration, we feel that it has merit but does not fully meet PLOS ONE’s publication criteria as it currently stands. Therefore, we invite you to submit a revised version of the manuscript that addresses the points raised during the review process. Please see the comments in comments section.

We look forward to receiving your revised manuscript.

Kind regards,

Krishna Kumar Aryal

Academic Editor

PLOS ONE

Journal Requirements:

Additional Editor Comments:

Revised title: the correct name of GSHS is Global School-based Student Health Survey. Kindly use the right one. Abbreviated form in title may not be required.

Abstract: The data source being GSHS is not explicitly mentioned. Mention that in the abstract.

Introduction: the line mentioning objective at the end of the introduction is replaced with the new title of the objective and this does not look appropriate. There was no need to change it this way. The previously mentioned objective was fine.

Methods:

GSHS is not provided with full form in its first use in the main body of the manuscript. Please review the manuscript thoroughly one more time, for any other such errors.

LDCs was mentioned as the country selection criteria, however, the countries listed in the study do not match the list as per the UN classification of LDCs (https://www.un.org/ohrlls/content/profiles-ldcs). Include citation for LDCs in the manuscript. A better option to select countries would be to follow the world bank classification of countries by income level. But it’s up to authors to decide.

In addition, the authors have provided clarification for non-inclusion of dataset beyond 2013 as non-availability of the data. But the datasets for some LDCs (some were verified by the editor) are available beyond 2013 as well (https://extranet.who.int/ncdsmicrodata/index.php/catalog/GSHS). The link provided by the authors is not the right one for dataset, that is for the questionnaires. Not sure from where you accessed the dataset. Include citation for the source of dataset in the manuscript. With this, authors are advised to add the newer datasets to the analysis which makes the manuscript more relevant to the present context. We believe you definitely have the codes of data cleaning and analysis stored; hence it should not take long time to reanalyze. Once you submit the revised version, your manuscript will be given top priority for review on time, considering the time consumed in rework of this important manuscript. Hence, authors are requested to re-consider the comment by reviewer 2 about the datasets by time.

Minor general comment: kindly do a thorough copy editing of the manuscript before resubmitting so that typos and other grammatical errors are minimized, and any issues with language and presentation are corrected.

All the best!

---

## [Author Response · Author response to Decision Letter 2]

29 Nov 2023

We appreciate the reviewers for your precious time in reviewing our paper and providing valuable comments. Your valuable and insightful comments led to possible improvements in the current version. The authors have carefully considered the comments and tried our best to address every one of them.

---

## [Editor Report · Decision Letter 3]

8 Dec 2023

PONE-D-23-03514R3The prevalence of alcohol consumption and its related factors in adolescents: findings from Global School-based Student Health SurveyPLOS ONE

Dear Dr. salemi,

Thank you for submitting your manuscript to PLOS ONE. After careful consideration, we feel that it has merit but does not fully meet PLOS ONE’s publication criteria as it currently stands. Therefore, we invite you to submit a revised version of the manuscript that addresses the points raised during the review process. Please see one minor comment and submit the revised version.

We look forward to receiving your revised manuscript.

Kind regards,

Krishna Kumar Aryal, MPH, PhD

Academic Editor

PLOS ONE

Journal Requirements:

Additional Editor Comments (if provided):

Dear Authors,

Thank you for considering the comments from reviewers and editor.

During the latest revision, you seem to have copy and pasted the whole title while trying to summarize the objective in the first para of discussion. Kindly check and correct.

And share the revised version.

Thank you!

---

## [Author Response · Author response to Decision Letter 3]

24 Dec 2023

Dear Editor, 

Thanks for your comment and we amended the your comment in discussion part. 

Additional Editor Comments (if provided):

Dear Authors,

Thank you for considering the comments from reviewers and editor.

During the latest revision, you seem to have copy and pasted the whole title while trying to summarize the objective in the first para of discussion. Kindly check and correct.

And share the revised version.

Thank you!

---

## [Editor Report · Decision Letter 4]

2 Jan 2024

The prevalence of alcohol consumption and its related factors in adolescents: findings from Global School-based Student Health Survey

PONE-D-23-03514R4

Dear Dr. salemi,

We’re pleased to inform you that your manuscript has been judged scientifically suitable for publication and will be formally accepted for publication once it meets all outstanding technical requirements.

Kind regards,

Krishna Kumar Aryal, MPH, PhD

Academic Editor

PLOS ONE
---

## [Editor Report · Acceptance letter]

20 Mar 2024

PONE-D-23-03514R4 

PLOS ONE

Dear Dr. salemi, 

I'm pleased to inform you that your manuscript has been deemed suitable for publication in PLOS ONE. Congratulations! Your manuscript is now being handed over to our production team.

Kind regards, 

on behalf of

Dr Krishna Kumar Aryal 

Academic Editor

PLOS ONE